# Intraspecific Fine-Root Trait-Environment Relationships across Interior Douglas-Fir Forests of Western Canada

**DOI:** 10.3390/plants8070199

**Published:** 2019-06-30

**Authors:** Camille E. Defrenne, M. Luke McCormack, W. Jean Roach, Shalom D. Addo-Danso, Suzanne W. Simard

**Affiliations:** 1Department of Forest and Conservation Sciences, Faculty of Forestry, University of British Columbia, Vancouver, BC V6T 1Z4, Canada; 2Center for Tree Science, The Morton Arboretum, Lisle, IL 60532, USA; 3Skyline Forestry Consultants Ltd., Kamloops, BC V2C 1A2, Canada; 4CSIR-Forestry Research Institute of Ghana, KNUST, P. O. Box 63, Kumasi, Ghana

**Keywords:** belowground, biogeographic gradient, Douglas-fir, fine root, mycorrhizal fungi, plant traits, root diameter, root economics, root tissue density

## Abstract

Variation in resource acquisition strategies enables plants to adapt to different environments and may partly determine their responses to climate change. However, little is known about how belowground plant traits vary across climate and soil gradients. Focusing on interior Douglas-fir (*Pseudotsuga menziesii* var. *glauca*) in western Canada, we tested whether fine-root traits relate to the environment at the intraspecific level. We quantified the variation in commonly measured functional root traits (morphological, chemical, and architectural traits) among the first three fine-root orders (i.e., absorptive fine roots) and across biogeographic gradients in climate and soil factors. Moderate but consistent trait-environment linkages occurred across populations of Douglas-fir, despite high levels of within-site variation. Shifts in morphological traits across regions were decoupled from those in chemical traits. Fine roots in colder/drier climates were characterized by a lower tissue density, higher specific area, larger diameter, and lower carbon-to-nitrogen ratio than those in warmer/wetter climates. Our results showed that Douglas-fir fine roots do not rely on adjustments in architectural traits to adapt rooting strategies in different environments. Intraspecific fine-root adjustments at the regional scale do not fit along a single axis of root economic strategy and are concordant with an increase in root acquisitive potential in colder/drier environments.

## 1. Introduction

Functional traits of fine roots and mycorrhizal symbionts have become central to understanding belowground acquisition strategies and plant responses to environmental change from local to global scales [1,2,3,4,5]. Recent syntheses of large-scale datasets have facilitated exploration of interspecific (i.e., among-species) fine-root functional trait variation. These studies have notably advanced our understanding of the fundamental constraints underlying fine-root trait variation (e.g., phylogeny and climate and growth form [4,6,7,8,9]). Importantly, studies across plant species have reported inconsistent evidence for a single root economic spectrum varying from more conservative roots with high structural investment to more cheaply constructed, acquisitive root types [5,10,11,12]. As a result, a multidimensional root trait framework has begun to emerge [13,14,15,16,17].

Thus far, most studies have focused on assessing fine-root functional trait variation among species, following the assumption that among species differences in root trait values are greater than those within species, as found in studies of aboveground traits [18,19,20]. However, within-species variation in fine-root functional traits can also be important, reflecting both heritable genetic variation [21,22] and phenotypic plasticity [3,7,23]. Intraspecific variation in fine-root trait expression is, therefore, likely to be an important driver of plant community assembly and may enable populations of plants to adjust to environmental conditions and changing climate [7,21]. Despite recent studies stressing the need for empirical research that links intraspecific plant trait variation to the environment [20,24,25], such investigations with respect to fine roots have only been conducted on relatively few species and in limited environmental contexts [3,21,22]. For instance, Zadworny et al. [21,22] found that fine-root traits of Scots pine (*Pinus sylvestris*) were related to mean annual temperature (MAT) as roots were thicker with lower specific root length (SRL) and lower root tissue density (RTD) in colder environments. The larger diameters were associated with a greater cortex area, which may indicate an overall shift among Scot pine trees from northern populations to adapt to cold environments by building cheaper fine roots (low RTD), with potentially higher absorptive capacity. Alternatively, across a similar temperate to boreal transect, Ostonen et al. [3] showed that Norway spruce (*Picea abies*) and Scots pine trees, at higher latitudes, had longer and thinner fine roots with higher RTD, compared to trees in warmer, lower latitude forests. According to Ostonen et al. [3], these adjustments were also closely related to an overall increase in absorptive root biomass. Despite some similarities, notable contradictions in the aforementioned studies demonstrate the need to better understand intraspecific belowground trait-environment linkages [1,2,6].

To test whether fine-root functional traits relate to the environment at the intraspecific level, we quantified root trait variation in interior Douglas-fir (*Pseudotsuga menziesii* var. *glauca* (Beissn). Franco; hereafter Douglas-fir) across a biogeographic gradient, in western Canada. In a previous study across the same gradient, we focused on variation in the ectomycorrhizal fungal species community and functional traits [26]. We notably found that fungi with rhizomorphs (e.g., *Piloderma* sp.) or proteolytic abilities (e.g., *Cortinarius* sp.) dominated communities in the warmer and less fertile environments of the gradient. Conversely, Ascomycetes (e.g., *Cenococcum geophilum*) or fungi that explore short distances in the soil were favored the in colder/drier environments where soils were richer in total nitrogen (N). This previous study was notably inconclusive regarding the potential link between root and fungal acquisition strategies at the regional scale.

In the present study, we measured aspects of root morphology including fine-root diameter, RTD, SRL and specific root area (SRA). We also assessed fine-root chemical (root N concentration and root carbon-to-nitrogen ratio, C:N) and architectural (branching intensity and dichotomous branching index, DBI) traits. We first hypothesized that MAT would be an important driver of root trait variation such that trade-offs among traits would favor greater root acquisitive capacity in colder climates in order to compensate for more limited resource availability and a shorter window for growth and acquisition. These trade-offs should translate into a higher SRL and SRA and lower root diameter, RTD, and root C:N because such fine roots are cheaply-constructed and are assumed to be more acquisitive, whereas thicker fine roots with a higher construction cost are hypothesized to be more conservative [13]. Increased root branching intensity, particularly within resource patches, is generally associated with greater root metabolism and enhanced root proliferation as it results in a higher density of more metabolically active, and more absorptive, first- and second-order roots [21,27,28]. Therefore, we also expected higher root branching intensity in colder/drier environments. We explored pairwise trait relationships to assess whether trait trade-offs, at the intraspecific level, are consistent with an acquisitive-conservative trait spectrum. We also partitioned variation in fine-root traits at different ecological scales, starting with the regional scale and then to the site, tree, and individual root branch levels, and tested a second hypothesis that the majority of the variation in fine-root traits of Douglas-fir would occur at the regional scale, consistent with aboveground traits [29].

## 2. Results

### 2.1. Fine-root Trait Response to Abiotic Factors

Responses of fine-root traits to abiotic factors were mixed and these factors only explained around 10% of the variation in fine-root traits, as most of the variation occurred at small ecological scales (e.g., root branch, individual soil block; Figure 1). With the exception of SRL, most traits were responsive to at least one abiotic factor, yet the direction and strength of these responses were often dependent on root order (Figure 2; Table 1 and Appendix A). Root diameter, SRA, and RTD each responded significantly to different aspects of climate. However, the size of the effect of environmental factors was relatively small (Figure 2), and the proportion of the variance explained by the different climatic factors was low (all marginal R^2^ < 0.1; Table 1). Still, most first- and second-order root morphological traits were significantly related to MAT, with root diameter and RTD increasing with MAT, while SRA decreased with MAT (*p* < 0.05). In contrast, to first- and second-order roots, the diameter of third-order roots decreased with mean annual precipitation (MAP) as well as cation exchange capacity (CEC), while SRA of third-order roots increased with soil available P. Root tissue density of third-order roots was unrelated to any abiotic factor (Figure 2). The C:N ratios of first- and third-order roots were most responsive to soil properties, with C:N of first- and third-order roots being positively related to soil C:N. The C:N ratio of third-order roots was also positively related to MAT, but it was negatively related to CEC and soil available P (Figure 2; Table 1). The positive relationship between branching intensity and soil available P was only marginally significant (marginal R^2^ = 0.02, *p* = 0.05; Table 1), while DBI (values range from 0, a fully dichotomous branching pattern to 1, a fully herringbone branching pattern [30,31]) was not related to any of the environmental factors considered in this study.

### 2.2. Ordination and Trait Correlation

Consistent among the three root orders, root trait variation did not fit into one dimension of the ordination (Figure 3 and Appendix A). Root morphological traits (SRL, SRA, diameter, and, to a lesser extent, RTD), were well correlated with the first dimension, which explained c. 30% of the variation for the three root orders and represented a gradient from thinner, high SRL roots to thicker, low SRL roots. This gradient was present within regions, as the five regions considered were not well separated along the first axis. The second axis of variation accounted for c. 22% of the total variation and was best represented by chemical traits (root C and N; root C:N was not included to avoid redundancy). Regions were well separated across this axis, which represented a gradient of higher root N (Kamloops, Revelstoke) to lower root N (Nelson, Williams Lake). Root architectural traits were not well represented along any of the axes (scores < 0.15, for each root order) nor were they well captured by the third PC axis which accounted for <20% of the variation in each root order Figure 3 and Appendix A).

Consistent within each root order, the variation in RTD was negatively correlated with that of root diameter, which was weakly negatively correlated with the variation in SRL (Appendix A). As expected from mathematical dependencies (i.e., the root volume and area depend on root diameter), variations in root diameter, SRA, and RTD were correlated. However, the variation in root C:N ratio was not correlated with the variation in morphological traits. Architectural traits were not correlated with any of the morphological traits.

### 2.3. Partition of Fine-root Trait Variance

More than half of the root morphological trait variance in Douglas-fir was expressed at small ecological scales (tree cluster and branch levels; Figure 1; Appendix A). In other words, differences among fine-root branches within individual soil blocks explained most of the variation in root diameter, RTD, SRL, and SRA. Consistent among morphological traits and root orders, the variation among soil blocks (within sites) accounted for, on average, 20% of the total variation. This pattern of high variance at small scales was even stronger for the architectural traits (branching intensity and DBI), which expressed over 90% of their variation among branches and tree cluster samplings (Appendix A). Alternatively, the proportion of the root trait variance at the cross-site level was negligible for root diameter, SRL and SRA (Figure 1). At the regional scale, intraspecific variation was the highest for root C:N and RTD. On average for the three orders, the regional scale accounted for roughly a quarter and 8% of the total variation in root C:N and RTD, respectively.

## 3. Discussion

Moderate but consistent trait-environment linkages occurred across populations of Douglas-fir that were distributed across climatic and edaphic gradients in western Canada, despite high levels of within-site variation. Our first hypothesis was partly confirmed, as MAT was the environmental variable that was most highly correlated with fine-root morphological traits. However, fine-root C:N was more responsive to soil properties (soil C:N, soil avail. P and CEC). Generally, fine roots in colder or drier climates were characterized by potentially higher acquisitive capacities (based on trade-offs among morphological and chemical traits), and variations in morphological and chemical traits represented two separate axes for fine-root adjustments. We rejected our second hypothesis as root trait variance was unevenly distributed across ecological scales, with over 50% of the variation in morphological traits occurring within individual soil blocks at a single site.

### 3.1. Fine-root Functional Traits Relates to the Environment

#### 3.1.1. Fine-root Morphology

Across the biogeographic gradient, first- and second-order roots of Douglas-fir trees tended to increase in diameter and SRA but decreased in RTD with decreasing MAT. This result partly agrees with our first hypothesis because, with the exception of increasing root diameter, these trait adjustments are expected to increase root resource acquisition potential, which was expected under colder climatic conditions. The response of Douglas-fir morphological and chemical traits to MAT were largely consistent with that of Scots pine absorptive fine roots reported by Zadworny et al. [22]. However, these results are in opposition to those reported by Ostonen et al. [3], despite similar temperature ranges used in both previous studies. To adjust to colder conditions, fine roots may increase their potential for soil resource acquisition to compensate for a more limited resource availability and a shorter window for growth and acquisition [14,15,16,28,32]. Alternatively, trees in colder environments may build fine roots with higher tissue protection and persistence (e.g., an increased number of phellem layers, increased phenolic compound content [22]). Our results provide evidence that in environments where temperature limits the availability of soil resources, fine roots increase their potential to acquire resources via morphological adjustments manifesting as greater surface area of roots per unit biomass invested (i.e., higher SRA).

In contrast to increases in SRA and reduced RTD, the larger diameter roots observed at lower MAT are generally associated with a more resource-conservative strategy. Plants with a conservative strategy are frequently expected to have fine roots with low SRL, large diameter, high RTD, low N concentration, low uptake capacity, low respiration rate and a long life span [12]. However, large diameter roots could also be associated with increased associations with mycorrhizal fungi [7,33]. Therefore, the increase in Douglas-fir root diameter in our study may be associated with enhanced root absorptive capacity if it coincides with an increased association with mycorrhizal fungi or greater hyphal growth [34].

In our study system, we did not observe significant changes in the ectomycorrhizal colonization rate across the gradient, which averaged c. 95% for all the sites (data not shown), consistent with other studies in this region [35,36]. The constrained responses of the ectomycorrhizal colonization rate observed here may be because a Douglas-fir has relatively thick fine roots, which are generally associated with higher, and sometimes more constant, levels of fungal colonization [27,37,38]. However, a single measure of colonization rate may less relevant than ectomycorrhizal community composition in Douglas-fir forests [39] and may be more relevant in arbuscular mycorrhizal plant species [7,33]. In a complementary study across the same biogeographic gradient, we showed that root diameter was not related to patterns of ectomycorrhizal fungal exploration type (see [40]) and that fine roots with high RTD and low C:N were more frequently colonized by ectomycorrhizal fungi with short emanating hyphae [26]. Whether association with ectomycorrhizal fungi that proliferate short emanating hyphae could lead to increased acquisitive capacities of thick, large-diameter Douglas-fir fine roots requires further research. Additional assessments of the fungal hyphal production rate and densities in soils are also needed to better assess associations with, and potential reliance of, Douglas-fir trees on their ectomycorrhizal partners across environmental gradients [34]. Finally, we cannot exclude that the trade-offs observed among morphological traits may not necessarily stem from an optimization of resource acquisition by either Douglas-fir trees or ectomycorrhizal fungi. For example, interspecific competition with tree neighbors, which was not quantified in our study, could have affected the fine-root functional traits of Douglas-fir trees.

#### 3.1.2. Fine Root Chemistry and Architecture

The fine-root C:N ratio was most responsive to soil properties (soil C:N, soil available phosphorus and CEC). Fine-root N concentration increased with increasing soil pH, CEC and the decreasing soil C:N ratio, which is potentially associated with greater nutrient uptake at the level of the individual fine root. The more fertile soils in our study area occurred in colder/drier regions and likely limited the diffusion rate of soil resources. Therefore, high root N concentrations with higher SRL, higher SRA, and lower RTD, which are generally associated with a shorter root lifespan and may represent a strategy to thrive where nutrient availability is heterogeneous and intermittent due to seasonality and soil freezing [4]. In the less fertile soils of our study area, the growth of Douglas-fir trees may not be limited by the low nutrient availability because of the high MAT and MAP. Accordingly, in these environments, we observed a more resource-conservative root strategy (higher RTD, C:N ratio and lower SRA). As suggested by Freschet et al. [6], in wetter environments with low nutrient availability, investments in higher branching intensity and/or mycorrhizal hyphae may be more beneficial to capture available N prior to leaching rather than investing in high metabolism (higher root N concentration).

Our results do not provide strong evidence that Douglas-fir fine roots rely heavily on adjustments in architectural traits. The relatively low values and narrow range of variation of root branching intensity are consistent with those of other ectomycorrhizal gymnosperm species considered by Tobner et al. [41] and Liese et al. [28]. These low values could be related to the consistently high rate of ectomycorrhizal colonization across our gradient, which suggests that local proliferation of fungal hyphae instead of increased fine-root branching may be the primary pathway to facilitate greater proliferation and exploitation of the soil environment.

### 3.2. Intraspecific Root Trait Variation

Contrary to our second hypothesis, the highest proportion of root trait variation was not at the regional scale. Though aspects of fine-root trait variation were significantly related to abiotic factors across regions, morphological, architectural, and, to a lesser extent, chemical traits, expressed the majority of their variation among root branches obtained from soil blocks within individual forest stands. Although our study was primarily designed to investigate fine-root trait-environment linkages at the regional scale, these findings demonstrate that processes at lower ecological scales are also important in determining root trait variation. It is not always feasible to intensively sample and quantify root trait variation at such small scales (i.e., within plot or even within tree/tree cluster variation), but in light of this result, care should be taken when interpreting and extrapolating a single mean value for a stand-level functional trait or for an individual species [42]. Thus, while environmental filters operate on the overall distribution of trait values within a region, their effects are lessened due to local variation among trees and root branches.

The high variation in root traits observed among branches within a single sampling location could be explained by differences in resource allocation or by differences in ectomycorrhizal symbiont identity. This may, in turn, affect carbon allocation to each root branch and the distinct morphology and chemistry expressed by individual roots [2,40]. For instance, the concentration of primary photosynthates in ectomycorrhizal root tips, such as starch, glucose, and non-structural carbohydrates, can change substantially among ectomycorrhizal symbionts [43]. Similarly, Pickles et al. [44] demonstrated that the distribution of many ectomycorrhizal individuals is often patchy. This leads to the possibility that different soil blocks from within the same site may be dominated by morphologically distinct ectomycorrhizas, contributing to the high variation in root traits at small spatial scales.

While our study design did not allow testing the relative contributions of genotypic vs. environmental variation, other work focused on aboveground traits in Douglas-fir as well as in root traits of Scots pine both suggest a high degree of genetic control on plant functional traits [21,45,46]. In this case, individuals and local populations of Douglas-fir may be limited in their ability to adjust to local changes in climate through phenotypic plasticity as root traits would primarily be under genetic controls. However, the high degree of within site variation observed here also indicates substantial within population root trait variation, which may enable some acclimation.

## 4. Conclusions

Across regional gradients of climate and edaphic factors in western Canada, the majority of Douglas-fir fine-root trait variance occurred within sites. However, we also identified moderate but consistent trait-environment linkages across populations of Douglas-fir. Generally, colder/drier climates were characterized by fine roots with a lower RTD, higher SRA, higher diameter, and lower C:N ratio. We also provided evidence for decoupled variations in fine-root morphological and chemical traits. These findings highlight the existence of multiple axes of within species fine-root adjustments that were consistent with increasing acquisitive potential of fine roots in harsher environments. The substantial within population root trait variation may then enable further acclimation at the stand level. As predicted changes in climate will likely impact belowground processes with important outcomes for tree persistence and resilience, further work connecting root traits and environmental variation will be particularly important to ensure that well-adapted tree populations are regenerated.

## 5. Materials and Methods

### 5.1. Biogeographic Gradient

A biogeographic gradient, including five study regions ranging in latitude from 49.6 to 51.7° N, was located within the natural range of Douglas-fir in British Columbia (Figure 4). Two regions (Kamloops and Williams Lake) were in the Interior Douglas-fir biogeoclimatic zone (IDF) and three regions (Salmon Arm, Nelson, and Revelstoke) were in the Interior Cedar-Hemlock biogeoclimatic zone (ICH) [47]. Regions were distributed along substantial precipitation and temperature gradients (see Table 1 in [26]; Appendix A). Sites in Williams Lake had the lowest MAT (on average 3.4 °C) followed by Revelstoke, Kamloops, Salmon Arm and Nelson (on average 7.3 °C). The driest region was Kamloops (average MAP, 441 mm), and the wettest region was Revelstoke (average MAP 1200 mm). Unlike other large environmental gradients that often correspond to wide ranges in latitude and daylength (e.g., [3]), the limited latitudinal range encompassed here corresponds to minimal differences in day length among the study regions.

In each region, three replicated study sites separated by at least 400 m were selected in naturally regenerated, mature, closed-canopy forest stands on ecosystems that best reflect the regional climate (namely, zonal ecosystems [47]). The average stand age at each region ranged from 98 years old (Revelstoke) to 143 years old (Salmon Arm; Appendix A). The northern-most stands in the IDF zone were growing on Luvisolic soils, and the southern-most stands in the ICH zone occurred predominantly on Brunisolic soils (Appendix A; [48,49]). Soils in the southern-most regions (Nelson, Revelstoke and Salmon Arm) were N-limited, but the mineral soil available P concentration was up to five times greater than that in the northern regions. The mineral soils in Revelstoke and Nelson were also characterized by low CEC and low soil pH compared to those in the northern regions (see Table 1 in [26]). In the semi-arid regions of interior British Columbia, Douglas-fir occurs in pure stands (in the IDF), while in the wetter regions, Douglas-fir trees grow in mixed species forests (in the ICH; [44]). Consequently, six sites contained pure Douglas-fir forest stands, and nine sites had mixed stands (Appendix A). The proportion of the Douglas-fir by basal area ranged from 49% in the mixed, evenly-aged forest stands of Salmon Arm to 100% in the pure, unevenly-aged forest stands of Kamloops (Appendix A).

### 5.2. Fine-Root Sampling and Processing

We used a nested strategy for sampling fine roots across ecological scales. A single sample plot of 30 m × 30 m containing at least ten dominant or co-dominant Douglas-fir trees was established at each site in summer 2016. We selected five healthy Douglas-fir trees per plot in a manner to avoid clumping of the sampling location. For each selected tree, a coarse root originating from the target tree was identified and traced 200 cm out from the trunk. At that point, a single soil block (20 cm × 20 cm × 20 cm) was extracted as closely as possible to the coarse root. Soil samples were collected by hammering a steel frame into the soil and the block were extracted using a flat shovel. In addition, one organic (L, F, and H layers) and one mineral soil (upper mineral horizons A and B, from the bottom of the organic layer to 10 cm depth) sample were collected using a trowel near the location of the soil block. Collected soil blocks and soil samples were stored individually in plastic bags, transported on ice to the laboratory within 1 to 4 days, and stored at 4 °C until processing (up to three months) to avoid alteration of fine-root traits that can occur with freezing. A total of 73 sample sets were collected in this study (5 regions, 3 sites per regions, 5 soil blocks per site, but only 3 blocks could be collected at Nelson site N2).

To extract fine roots, each soil block was soaked in water overnight, and washed over a 4 mm screen. All fine-root branches (diameter < 2 mm) and fragments >3 cm in length were recovered from the sieve and sorted by tree species. To do this, we developed a morphological key from root samples of known species identity collected from our study sites (Appendix A). The proportion of roots belonging to other tree species was not estimated. Douglas-fir roots that were turgescent with visible, intact periderm and that had (if present) colorful, swollen ectomycorrhizal tips were considered live roots. For trait measurements, we selected intact root branches containing at least three root orders, live ectomycorrhizal tips, and minimal breakage. Whenever possible, selected branches were carefully cleaned with a soft brush and tweezers and analyzed immediately after extraction. Otherwise, branches were stored in a plastic bag with a damp paper towel (changed daily) at 4 °C for no more than 3 days until further analysis.

### 5.3. Fine-Root Traits Measurements

We selected five, live and intact Douglas-fir fine-root branches from each of 73 soil blocks. A total of 365 root branches were scanned on a desktop scanner (400 dpi, 165 level gray scale, EPSON Perfection V800 Photo, STD 4800) and analyzed with WinRHIZO pro 2016 software (Regent Instruments Inc., Quebec City, Canada). Branches were analyzed for topology (magnitude, altitude, and external path length). We acknowledge the possible limitations of scanning roots at 400 dpi for root length measurements, particularly for very fine roots [50]. However, in our study system, this resolution was a good trade-off between speed and accuracy, as we avoided scanning overlapping roots (i.e., root length density < 1 cm cm^−2^). Furthermore, our scans had a good contrast between the roots and the background as Douglas-fir is a relatively thick-rooted tree species (first-order root diameter > 0.40 mm).

Following the initial scans of intact branches, each branch was divided into individual root orders using a scalpel under a stereomicroscope following the morphometric classification approach [51]. In our system, typical first-order roots were either comprised of ectomycorrhizal tips or displayed unbranched and uncolonized root tips. In all cases, we avoided thicker, longer pioneer roots [52]. Each root order group was scanned separately and analyzed for morphology (total length, total surface area, average diameter, and total volume). For the measure of root volume and area, we used the total value rather than the sum of values provided for each diameter class [53], because, in our study system, these two methods of estimation led to similar results (R^2^ = 0.99 for length and R^2^ = 0.97 for volume; data not shown). Root orders were then stored in envelopes, dried at 65 °C for 48 h, and weighed. For each root order group, SRL (m g^−1^) was calculated as the ratio of root length to root dry mass; SRA (cm^2^ g^−1^) as the ratio of root surface area to dry mass; and RTD (mg cm^−3^) as the ratio of root dry mass to root volume. To determine C and N concentration (%; Thermo Scientific Flash 2000 NC analyzer) in each of the three root orders, we randomly selected samples for each of the 15 sites as follows. Two soil blocks were randomly selected out of five originally sampled per site, and two root branches were randomly selected out of five originally sampled per block for a total of 180 root samples. The number of first-order roots for each branch was determined with the ImageJ software (National Institute of Health, USA). Root branching intensity was calculated as the number of first-order roots per length of second-order roots. The dichotomous branching index was calculated as:DBI = [Pe-min(Pe)]/[max(Pe)-min(Pe)],
with Pe, the external path length, defined as the sum of the number of root segments from the most distal root segment to the most basal root segment (i.e., third-order roots).

### 5.4. Climate and Soil Data

We obtained long-term averages for climatic variables over the period 1981–2010 from ClimateNA (http://www.climatewna.com/; [54]). To obtain soil properties, organic and mineral soil samples were air-dried and sieved to 2 mm. Samples were then sent to the analytical laboratory of the B.C. Ministry of Environment (Victoria, BC, Canada). Total soil C and N concentration (%) were measured using a combustion elemental analyzer (Thermo Scientific Flash 2000 NC analyzer). For available P. (PO_4_-P; orthophosphate as phosphorus), samples were prepared with the Bray P-1 method (dilute acid fluoride [55]) and analyzed with a UV/visible Spectrophotometer (Agilent Cary 60). To estimate the effective cation exchange capacity (CEC), cations were extracted from the soil samples with 0.1 M barium chloride [56] and analyzed with an ICP-OES spectrometer (Teledyne Leeman, Prodigy Dual view).

### 5.5. Data Analyses

Statistical analyses were conducted in R version 3.5.1 [57] and results were considered statistically significant at *p* < 0.05. To test whether fine-root functional traits related to the environment at the intraspecific level, we fitted linear mixed-effects models (LMMs). For each root order we separately considered SRL, SRA, RTD, root diameter, and root C:N, as response variables. For the third-order root C:N ratio, we used a linear model (multiple linear regression) instead, as the random effects were not significant. Models for branching intensity and DBI were fitted considering the whole absorptive root branch (as opposed to considering each root order separately). Before analyses, all response variables were log_10_-transformed to meet the assumptions of the LMMs. Data points that were >3 standard deviations from the region median were considered as statistical outliers and removed. This represented <2% of the data points for each trait and did not change the outcome of the LMMs. Climate (i.e., MAT and MAP) and soil variables (i.e., C:N ratio, avail. P, CEC) were added as fixed factors, while region, site, and tree were added as nested random factors. We also added stand age, Douglas-fir basal area, soil type, and stand composition (mixed vs. pure) as fixed factors. However, to avoid over-parametrization and multicollinearity among predictors, these variables were removed from LMMs as they all had a variance inflation factor >3 [58]. To fit LMMs, we used the ‘*lmer*’ function from the package *lme4* [59]. Global LMMs had this form:log_10_(Fine-root trait) ~ MAP + MAT + CEC + Soil C:N + Soil avail. P+ (1|region/site/tree).

We used the function ‘*step*’ from the *lmerTest* package to eliminate non-significant effects of LMMs based on the Akaike information criterion [60]. We checked LMMs adequacy with the ‘*plot_model*’ function from the *sjPlot* package [61] and LMMs fit using the conditional R^2^ (variance explained by the entire model, including both fixed and random effects) and marginal R^2^ (variance explained by the fixed effects) values, estimated following Nakagawa and Schielzeth [62]. We tested LMM significance with the log likelihood ratio and the significance of fixed effects with a type II Wald χ^2^ test. Standardized beta coefficients, and their 95% confidence intervals were extracted with the ‘*beta.coef*’ function from the *sjstats* package. Pairwise trait relationships were assessed using Spearman’s rank-order correlation because the assumption of pairwise linear relationships between variables of the Pearson product-moment correlation was violated. Trait coordination was explored using Principal Component Analysis (PCA). We acknowledge the mathematical dependence among root morphological traits and discuss the results accordingly. To quantify fine-root trait variation within sites, among sites, and across regions along the biogeographic gradient, we performed a variance partitioning analysis. For each root trait (i.e., SRL, SRA, RTD, root diameter, root C:N, Branching intensity, and DBI), we fitted linear models (ANOVA) with nested random effects in this order: region, site, tree. To partition variance among these hierarchically structured ecological scales, we used the function ‘*varcomp*’ from the package *ape* [60].

## Figures and Tables

**Figure 1 plants-08-00199-f001:**
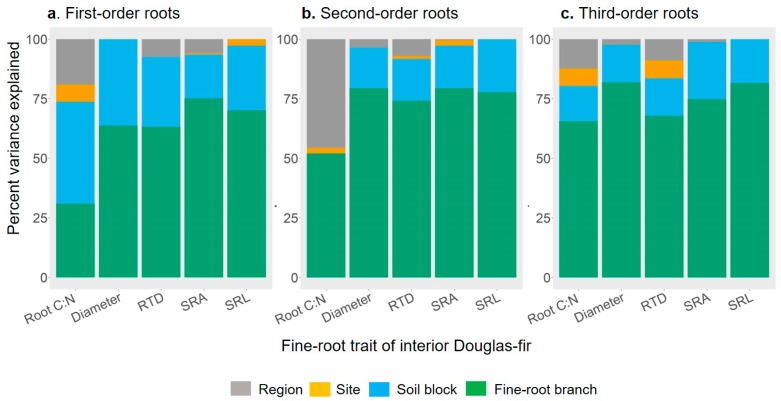
Variance partitioning of functional traits of the first three fine-root orders of interior Douglas-fir at different hierarchically structured ecological scales (region, site, soil block and fine-root branch). Root C:N, root carbon-to-nitrogen ratio; Diameter (mm); RTD, root tissue density (mg cm^−3^); SRA, specific root area (cm^2^ g^−1^); SRL, specific root length (m g^−1^).

**Figure 2 plants-08-00199-f002:**
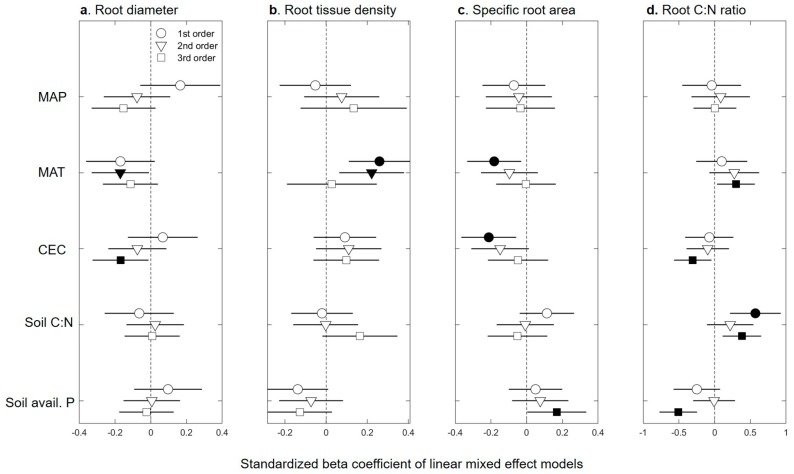
Effects of environmental variables (on the left of the figure) on interior Douglas-fir fine-root morphological (**a**–**c**) and chemical (**d**) traits. Standardized beta coefficients of linear mixed models (See Table 1) illustrate the effect of each environmental factor on a given fine-root trait in terms of its standardized effect size. MAP, mean annual precipitation (mm); MAT, mean annual temperature (°C); CEC, cation exchange capacity (cmol(+)kg^−1^); soil C:N, soil carbon-to-nitrogen ratio; soil avail. P, soil available phosphorus (ppm). Circles indicate average estimates and lines indicate 95% confidence intervals. Filled circles indicate a significant effect (*p* < 0.05) of a given environmental variable on a trait.

**Figure 3 plants-08-00199-f003:**
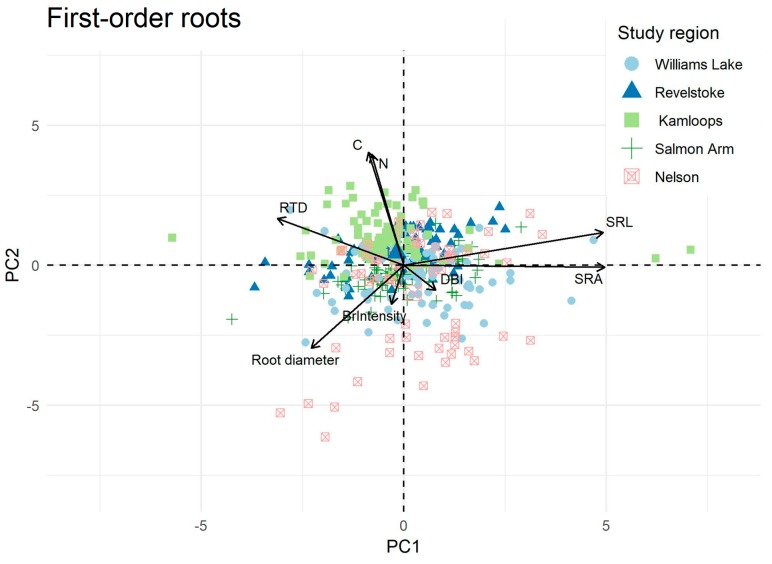
Ordination plot of study regions (five in total) across a biogeographic gradient based on principal component analysis of interior Douglas-fir first-order root traits. C, root carbon concentration (%); N, root nitrogen concentration (%); SRA, specific root area (cm^2^g^−1^); SRL, specific root length (mg^−1^); RTD, root tissue density (mgcm^−3^); BrIntensity, branching intensity (the number of first- order root/length of second- order root; cm^−1^); DBI, dichotomous branching index, values closer to 0 indicate a dichotomous branching pattern and values closer to 1, a herringbone branching pattern, see [31]. PC, principal component, PC1 = 30%; PC2 = 22%, see Appendix A.

**Figure 4 plants-08-00199-f004:**
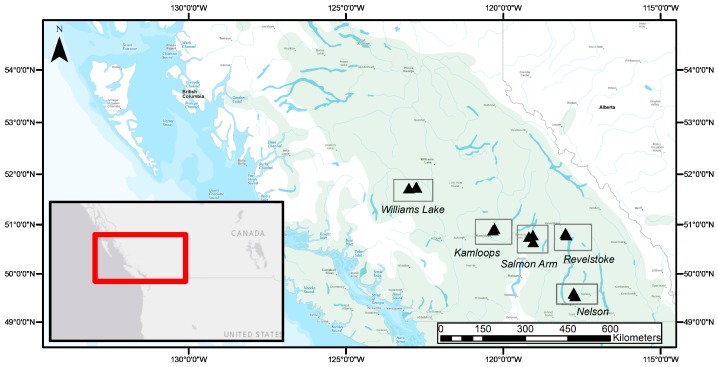
Geographical distribution of study regions (rectangles) and forest sites (3 triangles per study region) across the current natural range of interior Douglas-fir (*Pseudotsuga menziesii* var. *glauca*; green shading) in British Columbia, Canada. This figure was reproduced from [26].

**Table 1 plants-08-00199-t001:** Effect of climate (1980–2010) and soil variables on fine-root morphological, chemical, and architectural traits of interior Douglas-fir. Linear mixed-effects models were fitted with a restricted maximum likelihood. Model significance was tested with a log likelihood ratio test and significance of the fixed predictor(s) was tested with a type II Wald χ^2^ test. The specific root length and Dichotomous branching index were not included in the table because they were not related to any of the environmental factors considered in this study.

Data Trans.		LR ^1^ χ^2^	LR d.f.	*p*-Value	Model Fit		Predictor	Estimate	Standard Error	Wald χ^2^	*p*-Value
					Marginal R^2^	Conditional R^2^					
**Root diameter**											
1st	NA										
2nd	Log_10_	5.43	1	**0.02**	0.03	0.19	MAT ^2^	−0.01045	0.00397	7.16	**0.01**
3rd	Log_10_	7.50	2	**0.02**	0.03	0.18	MAP ^3^	−0.00007	0.00003	6.82	**0.01**
							CEC ^4^	−0.00173	0.00076	5.19	**0.02**
**Specific root area**											
1st	Log_10_	9.03	2	**0.01**	0.05	0.19	MAT	−0.00857	0.00572	5.56	**0.02**
							CEC	−0.00177	0.00059	7.33	**<0.01**
2nd	Log_10_	3.83	1	**0.05**	0.02	0.18	CEC	−0.00114	0.00056	4.10	**0.04**
3rd	Log_10_	4.81	1	**0.03**	0.02	0.22	Soil avail. P ^5^	0.00029	0.00006	5.03	**0.02**
**Root tissue density**											
1st	Log_10_	8.34	2	**0.01**	0.05	0.34	MAT	0.01826	0.00629	8.44	**<0.01**
							Soil avail. P	−0.00010	0.00005	3.98	**0.04**
2nd	Log_10_	5.61	1	**0.02**	0.04	0.20	MAT	0.01464	0.00560	6.82	**<0.01**
3rd	NA										
**Root C:N**											
1st	Log_10_	9.19	1	**<0.01**	0.28	0.60	Soil C:N ^6^	0.00854	0.00236	13.12	**<0.01**
2nd	NA										
3rd (multiple linear regression)	Log_10_	F-value = 7.17	4	**<0.01**	Multiple R^2^ = 0.35	Adjusted R^2^ = 0.30	MAT	0.02098	0.00965	4.72	**0.03**
							CEC	−0.00218	0.00087	6.3	**0.02**
							Soil C:N	0.00538	0.00173	9.66	**<0.01**
							Soil avail. P	−0.00028	0.00008	13.2	**<0.01**
**Root branching intensity**	Log_10_	3.97	1	0.05	0.02	0.18	Soil avail. P	0.00027	0.00013	4.07	**0.04**

Values in bold indicate statistically significant results at *p* < 0.05. data trans., data transformation; NA, not applicable. ^1^ LR, log likelihood ratio. ^2^ MAT, mean annual temperature (°C). ^3^ MAP, mean annual precipitation (mm). ^4^ CEC, effective cation exchange capacity (cmol(+)kg^−1^). ^5^ Soil avail. P, soil available phosphorus (PO_4_-P; orthophosphate as phosphorus; ppm). ^6^ Soil C:N, soil carbon-to-nitrogen ratio.

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
