# Peer review of "Intraspecific Fine-Root Trait-Environment Relationships across Interior Douglas-Fir Forests of Western Canada"

_plants, 2019, doi:10.3390/plants8070199_

Round 1
Reviewer 1 Report
Manuscript Review
Intraspecific fine-root trait-environment relationships across interior Douglas-fir forests of western Canada
By Camille Defrenne et al.
Submitted to Plants (Plants-530850)
This paper presents the results of an investigation on variation in morphological, chemical and architectural traits among the first three fine-root orders and across biogeographic gradients in climate and soil factors. The manuscript deserves publication after appropriate revisions.
L53-63. It is more suitable for the methods than for the introduction. It is better to revise it.
L78. References should be numbered properly; 28 must be before 29. This suggestion must be followed throughout the manuscript.
L79-87. Objectives should be written clearly.
L89. "abiotic filters" or "abiotic factors"?
L95. "effect sizes" is not clear.
L103-104. "while that ... also positively" should be rewritten.
L106-107. "[48-49]' after 29? References must be written in order.
L116-117. Not clear; should be rewritten.
L143. The correlation data must be shown here.
L162. What is weakly negatively correlation?
L231. Is it acceptable to cite unpublished data? It applies to other places as well.
L277. A clear conclusion is needed.
L387. It is better to use P<0.05 and smaller but not larger.
L406-422. References 57 and 59 are missing from the text.
L447. "thank to" or "thank"?
L454-610. Journal title should be abbreviated.
L605. Reference 57 is not cited.
L608. Reference 59 is not cited.
Table S2. Are these values "(-0.26, -0.20, -017, -0.23)" significant?
Author Response
Dear Reviewer,
We thank you a lot for you constructive and helpful review that will greatly improve our manuscript. Please see our point-by-point responses below (all of the corrections are in red in the revised Ms):
L53-63. It is more suitable for the methods than for the introduction. It is better to revise it.
We agree that the following sentence was more suitable for the methods and thus, we removed it from the introduction:" We related variation in fine-root traits to climate (mean annual temperature, MAT and mean annual precipitation, MAP; over the period 1981-2010) and edaphic (soil C:N, cation exchange capacity, CEC, and available phosphorus, avail. P) variables."
We kept our objective which is "To test whether fine-root functional traits relate to the environment at the intraspecific level, we quantified root trait variation in interior Douglas-fir (Pseudotsuga menziesii var. glauca (Beissn). Franco; hereafter Douglas-fir) across a biogeographic gradient, in western Canada."
We think that we need to mention our previous study on mycorrhizal fungi in the introduction to better contextualize this present study (We added a paragraph as requested by the second reviewer). We also introduce the root traits we measured so the hypotheses make more sense. Therefore, we chose to keep the following paragraphs in the introduction:
-->"In a previous study across the same gradient, we focused on variation in the ectomycorrhizal fungal species community and functional traits [26] [...]
-->"In the present study, we measured aspects of root morphology including fine-root diameter, root tissue density (RTD), specific root length (SRL) and specific root area (SRA). We also assessed fine-root chemical (root nitrogen, N, concentration and root carbon-to-nitrogen ratio, C:N) and architectural (branching intensity and dichotomous branching index, DBI) traits."
In order to improve our introduction, we also added the following paragraph:
"This is an important shortcoming when considering the potential outcomes of environmental change on plant populations. Furthermore, identifying meaningful patterns of intraspecific fine-root trait variation may give insights into structural investments in fine roots in relation to their local environment. For instance, Zadworny et al. (2016, 2017) found that fine-root traits of Scots pine (Pinus sylvestris) were related to mean annual temperature (MAT) as roots were thicker with lower specific root length (SRL) and lower root tissue density (RTD) in colder environments. The larger diameters were associated with greater cortex area, which may indicate an overall shift among Scot pine trees from northern populations to adapt to cold environments by building cheaper fine roots (low RTD), with potentially higher absorptive capacity. Alternatively, across a similar temperate to boreal transect, Ostonen et al. (2017) showed that Norway spruce (Picea abies) and Scots pine trees, at higher latitudes, had longer and thinner fine roots with higher RTD, compared to trees in warmer, lower latitude forests. According to Ostonen et al. (2017), these adjustments were also closely related to an overall increase in absorptive root biomass. Despite some similarities, notable contradictions in the aforementioned studies demonstrate the need to better understand intraspecific belowground trait-environment linkages (Freschet et al. 2017; Laliberté 2017; McCormack et al. 2017)."
L78. References should be numbered properly; 28 must be before 29. This suggestion must be followed throughout the manuscript.
Thanks for this comments, we updated the reference list.
L79-87. Objectives should be written clearly.
Our objective is "To test whether fine-root functional traits relate to the environment at the intraspecific level, we quantified root trait variation in interior Douglas-fir (Pseudotsuga menziesii var. glauca (Beissn). Franco; hereafter Douglas-fir) across a biogeographic gradient, in western Canada."
We removed this following paragraph because we feel that it confuses the reader: "Here, across gradients of climate and edaphic factors, moderate but consistent trait-environment linkages occurred across populations of Douglas-fir, despite high levels of within-site variation. We highlight the existence of multiple axes of within-species fine-root adjustments that were consistent with greater acquisitive potential of fine roots in colder/drier than warmer/wetter environments. The substantial within-population root trait variation may then enable further acclimation to changing environmental conditions. Together, these results advance our functional understanding of intraspecific fine-root adjustments to the environment which is crucial to helping us ensure regeneration of tree populations that are well-adapted to their environmental conditions."
L89. "abiotic filters" or "abiotic factors"?
We changed it for "abiotic factors" for consistency
L95. "effect sizes" is not clear.
We changed to: "However, the size of the effects of environmental factors was relatively small"
L103-104. "while that ... also positively" should be rewritten.
We changed to : "The C:N ratio of third-order roots was also positively related to MAT but it was negatively related to CEC and soil available P (Figure 2; Table 1)."
L106-107. "[48-49]' after 29? References must be written in order.
We addressed this by updating the reference list
L116-117. Not clear; should be rewritten.
We changed to: "Effects of environmental variables (on the left of the figure) on interior Douglas-fir fine-root morphological (a; b and c) and chemical (d) traits"
L143. The correlation data must be shown here.
We apologize but we do not understand this comment. Do you mean that we need to bring the correlation table (Table S2) into the main text?
L162. What is weakly negatively correlation?
We changed to : "Consistent within each root order, the variation in RTD was negatively correlated with that of root diameter and it was weakly negatively correlated with the variation in SRL "
L231. Is it acceptable to cite unpublished data? It applies to other places as well.
Yes, you can refer to Plants "Instruction for Authors" > References:
Unpublished work, submitted work, personal communication:
Author 1, A.B.; Author 2, C. Title of Unpublished Work. status (unpublished; manuscript in preparation).
L277. A clear conclusion is needed.
We added a conclusion.
L387. It is better to use P<0.05 and smaller but not larger.
We removed " and marginally statistically significant at P< 0.1."
L406-422. References 57 and 59 are missing from the text.
We addressed this by updating the reference list
L447. "thank to" or "thank"?
We replaced it for "Thank to"
L454-610. Journal title should be abbreviated.
We abbreviated the journal titles.
L605. Reference 57 is not cited.
We addressed this by updating the reference list
L608. Reference 59 is not cited.
We addressed this by updating the reference list
The last reference (#63) is cited in the supplementary materials
Table S2. Are these values "(-0.26, -0.20, -017, -0.23)" significant?
Yes these values are significant. We used the 'cor.test' function in R, please see the results below:
SRL vs RTD (-0.26)
S = 10245000, p-value = 3.043e-07
SRA vs D (-0.20)
S = 9743000, p-value = 0.0001004
RTD vs SRL (-0.17)
S = 9446100, p-value = 0.001505
D vs RTD (-0.23)
S = 10004000, p-value = 6.041e-06

Reviewer 2 Report
This is a carefully executed study that addresses an important question in functional traits of vascular plants. While reading it, I wondered whether the authors were considering the role of the fungi in the nutrient acquisition system of Douglas-fir. Upon reaching the discussion, it became apparent that there is extensive information available about the role of the fungi. I think it would be helpful if some of that information could be placed in the introduction, so that the results for the root component could be seen in a broader context. Ideally, it would be helpful if the variation in root traits could be compared to the variation in fungal species or traits. The surprising lack of variation found at levels above the individual branch may not be found when characteristics of the associated fungi are considered.
The quality of writing and presentation is very high. I found one typo.
l. 253 gymnosperms should be singular
Author Response
Dear Reviewer,
We thank you a lot for you constructive and helpful review that will greatly improve our manuscript. Please see our point-by-point responses below (all of the corrections are in red in the revised Ms):
I think it would be helpful if some of that information could be placed in the introduction, so that the results for the root component could be seen in a broader context.
Thanks for this comment. We added the following paragraph in the introduction: "In a previous study across the same gradient, we focused on variation in the ectomycorrhizal fungal species community and functional traits [26]. We notably found that fungi with rhizomorphs (e.g., Piloderma sp.) or proteolytic abilities (e.g., Cortinarius sp.) dominated communities in the warmer and less fertile environments of the gradient. Conversely, Ascomycetes (e.g., Cenococcum geophilum) or fungi that explore short distances in the soil were favored the in colder/drier environments where soils were richer in total nitrogen (N). This previous study was notably inconclusive regarding the potential link between root and fungal resource foraging strategies at the regional scale."
In order to improve our introduction, we also added the following paragraph:"This is an important shortcoming when considering the potential outcomes of environmental change on plant populations. Furthermore, identifying meaningful patterns of intraspecific fine-root trait variation may give insights into structural investments in fine roots in relation to their local environment. For instance, Zadworny et al. (2016, 2017) found that fine-root traits of Scots pine (Pinus sylvestris) were related to mean annual temperature (MAT) as roots were thicker with lower specific root length (SRL) and lower root tissue density (RTD) in colder environments. The larger diameters were associated with greater cortex area, which may indicate an overall shift among Scot pine trees from northern populations to adapt to cold environments by building cheaper fine roots (low RTD), with potentially higher absorptive capacity. Alternatively, across a similar temperate to boreal transect, Ostonen et al. (2017) showed that Norway spruce (Picea abies) and Scots pine trees, at higher latitudes, had longer and thinner fine roots with higher RTD, compared to trees in warmer, lower latitude forests. According to Ostonen et al. (2017), these adjustments were also closely related to an overall increase in absorptive root biomass. Despite some similarities, notable contradictions in the aforementioned studies demonstrate the need to better understand intraspecific belowground trait-environment linkages (Freschet et al. 2017; Laliberté 2017; McCormack et al. 2017)."
Ideally, it would be helpful if the variation in root traits could be compared to the variation in fungal species or traits.
We touched on the comparison between fine-root and fungal traits in our other paper.
The surprising lack of variation found at levels above the individual branch may not be found when characteristics of the associated fungi are considered.
We agree with your statement. We added the following paragraph in the discussion: "The high variation in root traits observed among branches within a single sampling location could be explained by differences in resource allocation or by differences in ectomycorrhizal symbiont identity. This may in turn affect carbon allocation to each root branch and the distinct morphology and chemistry expressed by individual roots [2,40]. For instance, the concentration of primary photosynthates in ectomycorrhizal root tips such as starch, glucose and non-structural carbohydrates, can change substantially among ectomycorrhizal symbiont [43] . Similarly, Pickles et al. [44] demonstrated that the distribution of many ectomycorrhizal individuals is often patchy. This leads to the possibility that different soil blocks from within the same site may be dominated by morphologically distinct ectomycorrhizas, contributing to the high variation in root traits at small spatial scales."
253 gymnosperms should be singular
We corrected this typo, thanks

Round 2
Reviewer 2 Report
The added text helps to place the results in a wider context.